# Migration of Dihydroxy Alkylamines and Their Possible Impurities from Packaging into Foods and Food Simulants: Analysis and Safety Evaluation

**DOI:** 10.3390/polym15122656

**Published:** 2023-06-12

**Authors:** Antía Lestido-Cardama, Letricia Barbosa-Pereira, Raquel Sendón, Perfecto Paseiro Losada, Ana Rodríguez Bernaldo de Quirós

**Affiliations:** 1Department of Analytical Chemistry, Nutrition and Food Science, Faculty of Pharmacy, University of Santiago de Compostela, 15782 Santiago de Compostela, Spain; antia.lestido@usc.es (A.L.-C.); letricia.barbosa.pereira@usc.es (L.B.-P.); raquel.sendon@usc.es (R.S.); perfecto.paseiro@usc.es (P.P.L.); 2Instituto de Materiales (iMATUS), University of Santiago de Compostela, 15782 Santiago de Compostela, Spain

**Keywords:** dihydroxyalkylamines, impurities, polypropylene, LC-MS/MS, target and non-target methods, exposure

## Abstract

Alkyl diethanolamines are a group of compounds commonly used as antistatic agents in plastic food packaging materials. These additives and their possible impurities have the ability to transfer into the food; hence, the consumer may be exposed to these chemicals. Recently, scientific evidence of unknown adverse effects associated with these compounds was reported. N,N-bis(2-hydroxyethyl)alkyl (C8-C18) amines as well as other related compounds and their possible impurities were analyzed in different plastic packaging materials and coffee capsules using target and non-target LC-MS methods. N,N-bis(2-hydroxyethyl)alkyl amines, precisely, C12, C13, C14, C15, C16, C17 and C18, 2-(octadecylamino)ethanol and octadecylamine, among others, were identified in most of the analyzed samples. It should be emphasized that the latter compounds are not listed in the European Regulation 10/2011 and 2-(octadecylamino)ethanol was classified as high toxicity according to Cramer rules. Migration tests were carried out in foods and in the food simulants Tenax and 20% ethanol (*v*/*v*). The results showed that stearyldiethanolamine migrated into the tomato, salty biscuits, salad and Tenax. Lastly, as a crucial step in the risk assessment process, the dietary exposure to stearyldiethanolamine transferred from the food packaging into the food was determined. The estimated values ranged from 0.0005 to 0.0026 µg/kg bw/day.

## 1. Introduction

Plastics are still the most widely used materials for food packaging. In Europe, packaging applications represent the largest portion of end-use materials, within 40.5% of the total plastics demand in 2020 [1]. Some examples of the most common plastic materials used in food packaging applications are polyethylene (PE), polypropylene (PP), and polyethylene terephthalate (PET) [2]. In order to improve their functional properties and modify their characteristics according to their final use, different additives are incorporated. Antioxidants, slip agents, plasticizers, lubricants, light stabilizers, and antistatic agents are some of the most common additives used in food packaging applications [3,4]. Antistatic agents are used to avoid a build-up of static electric charge on the surface of the plastic materials [3]. Alkyl diethanolamines are a group of compounds used as antistatic agents [5]. These alkylamines can be ethoxylated in a further process step to increase their water solubility [6]. Ethoxylated amines, together with glycerol monostearate, represent more than 50% of the antistatic market. Ethoxylated amines are more suitable for polyolefins than glycerol monostearate as they migrate slower to the surface of the polymer than glycerol monostearate. Generally, the preferred option is the combination of both antistatic agents that allows immediate and long-term antistatic protection due to the synergistic effect between the two classes of compounds [7]. In particular, ethoxylated fatty amines are extensively used as non-ionic antistatic agents [8]. Tallow is the traditional raw material used to produce fatty amines [9]. For example, tallow bis(2-hydroxyethyl)amine has been used in different polyolefins such as PE, PP, low-density polyethylene (PE-LD), high-density polyethylene (PE-HD) and linear low-density polyethylene (PE-LLD) [10].

Currently, additives such as N,N-bis(2-hydroxyethyl)alkyl (C8-C18) amine and N,N-bis(2-hydroxyethyl)alkyl (C8-C18) amine hydrochlorides are permitted for use in plastic food contact materials. The total specific migration limit (SML(T)) for these additives is set at 1.2 mg/kg of food, as outlined in Regulation 10/2011 [11].

Recently, in light of new scientific evidence on unknown adverse effects of N,N-bis(2-hydroxyethyl)alkyl (C8-C18) amine hydrochlorides, and given the potential concerns related to these substances, the European Food Safety Authority (EFSA) launched a call to collect data and information on the characterization and safety of N,N-bis(2-hydroxyethyl)alkyl (C8-C18) amine and N,N-bis(2-hydroxyethyl)alkyl (C8-C18) amine hydrochlorides for their use in plastic food contact materials and articles, following the European Commission request. The toxicity studies were carried out according to the OECD (Organisation for Economic Cooperation and Development) 414 test for prenatal developmental toxicity. In this context, the EFSA’s panel on Food Contact Materials, Enzymes, and Processing Aids (CEP) recently published a scientific opinion stating that there are no safety concerns for consumers when N,N-bis(2-hydroxyethyl)stearylamine, either partially or fully esterified with saturated C16/C18 fatty acids, is used in polymers intended to come into contact with dry foods at a maximum concentration of 2% (*w*/*w*) [12]. The information of interest sought includes migration data of substances and their impurities, reaction and degradation products, potential toxicity, impurities related to the manufacturing process, etc. [13].

The determination of dihydroxy alkylamines is generally accomplished with liquid chromatography coupled with mass spectrometry (LC-MS). For example, Vera et al. [14] identified several dihydroxy alkylamines between C8 and C22 in PP films using UPLC-MS/QTOF. Otoukesh et al. [5] studied the migration of dihydroxy alkylamines from PP coffee capsules into Tenax and coffee by employing UHPLC-MS/MS. In a recent work published by Vera et al. [15], ion mobility QTOF mass spectrometry has proven to be a powerful tool to analyze these types of compounds in PE.

In the present paper, alkylamines and other potential migrants were investigated in plastic food packaging materials. For that purpose, the LC-MS targeted method to determine N,N-Bis(2-hydroxyethyl)dodecylamine, stearyldiethanolamine, 2-(octadecylamino)ethanol and octadecylamine was carried out. Moreover, a non-targeted approach to detect and identify any component that can potentially migrate into food, including intentionally added substances called IAS (e.g., additives, monomers, etc.) and non-intentionally added substances called NIAS (e.g., impurities, reaction products, etc.), was applied. Special attention was paid to NIAS. The migration into food simulants and foods was also evaluated.

This study aims to contribute to providing new migration data of dihydroxyalkylamines in different food items and food simulants, as well as the identification of possible impurities and other components associated with these amines. In addition, dietary exposure data are also reported. These data can help the regulatory agencies to review the safety assessment of these substances for their use in plastic food contact materials.

## 2. Materials and Methods

### 2.1. Reagents and Chemicals

Ethanol absolute for analysis (EtOH), methanol LC-MS Grade (MeOH) and acetone analytical grade were provided from Merck (Darmstadt, Germany). Acetonitrile (ACN) LC-MS grade was obtained from Scharlau (Barcelona, Spain). Formic acid LC-MS Grade was supplied by Sigma-Aldrich (St. Louis, MO, USA). Ultrapure water was produced by an Autwomatic Plus purification system (Wasserlab, Navarra, Spain).

N,N-Bis(2-hydroxyethyl)dodecylamine (CAS 1541-67-9), 2-(octadecylamino)ethanol (CAS 31314-15-5) and octadecylamine ≥ 99% (CAS 124-30-1) were obtained from Sigma-Aldrich (St. Louis, MO, USA). Stearyldiethanolamine > 98% (CAS 10213-78-2) was obtained from Tokyo Chemical Industry Co., Ltd. (TCI; Tokyo, Japan). PEG-2 hydrogenated tallow amine (CAS 61791-26-2) was supplied by Toronto Research Chemicals (Toronto, ON, Canada). Tenax^®^ porous polymer adsorbent matrix 60-80 mesh was obtained from Supelco (St. Louis, MO, USA). QuEChERS Extract Pouch was supplied by Agilent Technologies (Sta. Clara, CA, USA). Charcoal activated (CAS 7440-44-0) was provided from Merck.

Stock solutions were prepared by dissolving the standards in EtOH with a concentration of 1000 mg/L, except for 2-(octadecylamino)ethanol, which was prepared at 250 mg/L. An intermediate mix solution of 10 mg/L was also prepared in EtOH. Calibration solutions ranging from 0.005 to 1 mg/L were then prepared by serial dilution from the intermediate mix solution using EtOH as the solvent. All solutions were stored in brown glass bottles at a temperature of 4 °C.

### 2.2. Samples

Twelve food-packaging samples were included in this study. Five samples were provided by the industry before the food or beverage contact, including packaging of Gouda cheese, orange juice, alcoholic beverage, surimi, and coffee capsules. The other seven samples were collected from packed foods purchased in a local supermarket (Santiago de Compostela, Galicia, Spain), including biscuits (sweet and salty), fresh pasta, banana, salad, and tomato. The packaging materials used for all food samples were made of plastic. The thickness of these packaging materials was measured three times using a Digimatic Micrometer from Mitutoyo, Japan. For more detailed information on the samples, please refer to Table 1.

In addition, PEG-2 hydrogenated tallow amine was also analyzed to check possible impurities. This surfactant, derived from animal fats, has been used in different applications. PEG compounds have been applied in several fields [16].

### 2.3. Sample Preparation: Packaging

Two different solvents were tested for the extraction, ACN and EtOH. Finally, the packaging materials were extracted using EtOH due to the high solubility of these compounds in this solvent. An extract of the packaging was obtained by immersion of a known surface area of the material (five pieces of 1 × 8 cm, 0.4 dm^2^) in 10 mL of the solvent and stored in an oven at 70 °C for 24 h. Each sample was extracted in duplicate.

### 2.4. Sample Preparation: Foodstuff

Samples were homogenized with a grinder or crushed to small sizes using a kitchen knife. Several extraction methods were investigated using a sweet biscuit sample (S7) and salad (S12). Finally, the selected extraction conditions are as follows: First, 5 g of sample were weighted for analysis and prepared in triplicate. Then, 10 mL of EtOH was added to samples for extraction and stirred for 10 min with a vibrant mixer (IKA^®^ Vibrax VXR basic, Germany). Then, the mixture was centrifuged at 5000 rpm for 10 min at 4 °C, and the supernatant was collected and filtered for injection into the liquid chromatograph. In the case of salad and fresh pasta (with ricotta and spinach), extracts were diluted by half with water (1:1, *v*/*v*) due to the intense green coloration. In order to monitor and control background contamination from the chemical compounds, an intermediate blank was included within each sample batch.

To carry out recovery tests, the sample of sweet biscuit (S7) and banana (S10) were selected. For this, samples were spiked at three different concentrations (0.02, 0.1 and 0.2 µg/g), letting it stand for 15 min before extraction. Then, the samples were extracted with the previously explained method.

### 2.5. Migration Tests

Migration tests were carried out on those samples whose packaging turned out to be positive and was not in previous contact with food using food simulants. The experimental conditions were selected in accordance with the expected conditions of use. Each migration was carried out in duplicate.

In the case of surimi packaging, the migration test took place in a migration cell by putting into contact the inner face with 30 mL of EtOH 20% (*v*/*v*) during 10 days in an oven at 40 °C. The contact surface was 0.68 dm^2^.

For the migration assay of the coffee capsules, Tenax^®^ was used as a dry food simulant since the capsules were designed to come into contact with this type of food. Prior to the migration test, Tenax^®^ was cleaned using acetone through Soxhlet extraction for a duration of 6 h. Subsequently, it was activated using heating, in an oven at 160 °C for 6 h. Following Regulation UNE-EN 14338, which specifies a ratio of 4 g of Tenax^®^ per square decimeter (dm^2^), the capsules were covered with 0.76 g of Tenax^®^ for the migration test [17]. After wrapping the capsules with aluminum foil, they were placed in an oven at 60 °C for a duration of 10 days. Following this period, the extraction of Tenax^®^ was carried out using 3 mL of EtOH, using the same extraction method employed for the food samples.

### 2.6. Instrumental Analysis

#### 2.6.1. Fourier Transform Infrared Spectroscopy (FTIR)

The identification of the packaging material was conducted using a Fourier Transform Infrared Spectroscopy with an Attenuated Total Reflectance (FTIR-ATR) from Jasco (Tokyo, Japan). Detailed information is reported elsewhere [18].

#### 2.6.2. Liquid Chromatography Coupled with Tandem Mass Spectrometry (LC-MS/MS)

The chromatographic system used in the analysis was from Thermo Fisher Scientific (San José, CA, USA). Detailed information is reported elsewhere [18].

The chromatographic separation was performed on a Kinetex XB.C18 100 Å (100 × 2.10 mm, 1.7 μm particle size) column from Phenomenex, thermostated at 40 °C. Water and MeOH were used as mobile phase A and B, respectively, with both solvents containing 0.1% formic acid. The flow rate and the injection volume were 300 µL/min and 10 µL, respectively. The gradient elution conditions used in the analysis were as follows: Initially, an isocratic mode with 30% solvent B is set for 1 min, followed by a gradient elution to 100% solvent B over 4 min. Subsequently, an isocratic elution with 100% organic phase was maintained for 13 min. Finally, the method returned to the initial conditions (30% B) at minute 27. A delay time of 3 min was set before recording the next chromatogram.

The mass spectrometer was operated in the positive electrospray ionization (ESI) mode. The operational parameters were as follows: The spray voltage was 3000 V, vaporizer temperature was set to 340 °C, and capillary temperature was maintained at 350 °C. Nitrogen gas served as both the sheath gas, operating at a pressure of 35 psi, and the auxiliary gas, operating at a pressure of 10 arbitrary units. Additionally, argon gas was utilized as the collision gas, exerting a pressure of 1.5 mTorr.

Data acquisition was performed in the selected reaction monitoring (SRM) mode. Two transitions for each compound were selected to facilitate identification, and the collision energy for each transition was optimized to maximize intensity. The mass spectrometry conditions can be found in Table 2. In addition, a non-targeted analysis to find other possible additives, components or possible impurities from these materials was carried out, acquiring data in full scan mode using the range of 40 to 400 *m*/*z*.

### 2.7. Quality Assurance/Quality Control

Validation parameters, such as sensitivity, linearity, intermediate precision, and recoveries were evaluated.

To verify the linearity of the detector response, standard solutions in EtOH were prepared at seven different concentration levels spanning the calibration range: 0.005, 0.01, 0.05, 0.1, 0.25, 0.5, and 1 mg/L. Each concentration level was analyzed in triplicate to ensure accuracy and consistency.

The sensitivity of the method was evaluated by determining the limit of detection (LOD) and limit of quantification (LOQ) for each individual compound in the standard solutions. The LOD was defined as a signal-to-noise ratio of 3, while the LOQ was defined as a signal-to-noise ratio of 10. These parameters were determined by considering the noise observed in the chromatographic analyses.

To assess intermediate precision and recoveries, the sample of sweet biscuit (S7) and banana (S10) were spiked with known amounts of the analytes at three different concentration levels (0.02, 0.1, and 0.2 µg/g). This process was performed in duplicate on three separate days, resulting in a total of six measurements. Sweet biscuit (S7) and banana (S10) samples were chosen as representative samples of cereals and fruits/vegetables, respectively.

## 3. Results and Discussion

### 3.1. Identification of the Packaging Material by FTIR-ATR

The results of the identifications are shown in Table 1. As can be seen, polypropylene (PP) was the most common polymer identified in the samples analyzed. Polypropylene exhibits distinctive peaks at various wavenumbers, including 2951 cm^−1^, 2873 cm^−1^, 1455 cm^−1^, 1375 cm^−1^, 1165 cm^−1^, 999 cm^−1^, and 843 cm^−1^ [19]. It makes sense since PP is one of the most common non-polar polymers used in the manufacturing of food contact materials due to its food functionality and relatively low cost [20]. Dihydroxy alkylamines are a class of tertiary amines commonly employed as antistatic agents. It has been observed that these compounds can migrate from various food contact materials, including polypropylene, polyethylene, polystyrene, and polyvinyl chloride [5]. Figure 1 illustrates the infrared (IR) spectrum comparison between the internal surface of sample S4 (represented by the black line) and the first entry of the library (represented by the red line), which corresponds to polypropylene.

### 3.2. Extraction Method Optimization

On the one hand, extraction solvents such as ACN and EtOH were tested to select the most suitable solvent for the packaging samples analyzed in this study, obtaining a greater number of peaks and greater intensity in the chromatograms from the extraction with EtOH.

On the other hand, in relation to food extraction, several methods were tested, with this aim; sweet biscuit (S7) and salad (S12) samples spiked at 0.2 µg/g were chosen to perform the assays. The determinations were performed in duplicate. Several extraction procedures were tested as follows: extraction of 5 g of food sample with (a) 10 mL of EtOH and stirring for 10 min; (b) 10 mL of EtOH and a set temperature (70 °C for 24 h), and (c) 10 mL of ACN and a sack of salts prepared for extraction using QuEChERS (Quick, Easy, Cheap, Effective, Rugged and Safe). With the three procedures evaluated, all the recoveries were in the acceptable range of 90 to 125%. Therefore, due to the speed and saving of reagents, the method selected consisted of extraction with EtOH and stirring for 10 min, followed by centrifugation. In the case of salad and fresh pasta (S9, S12), due to the intense green coloration of the extract, an attempt was made to clarify the extracts using charcoal activated, but the analytes of interest also precipitated, so it had to be diluted by half with water (1:1, *v*/*v*) before the chromatographic analysis.

### 3.3. Method Validation using LC-MS/MS

Quantification in this study was conducted using an external calibration curve. To establish the calibration curve, several calibration solutions with known concentrations (from 0.005 to 1 mg/L) were prepared in ethanol and analyzed during each working session. The chromatographic peaks’ areas in the selected transitions for quantification (as listed in Table 2) were plotted against the corresponding concentrations of the standard solutions to generate the calibration graphs. The fragmentation pattern obtained for N,N-Bis(2-hydroxyethyl) dodecylamine was compared to the pattern described in the study by Vera et al. [15] using the UNIFI software. This comparison confirmed the accurate identification of the compound. The linearity of the calibration function was evaluated, and the results are presented in Table 3. The determination coefficients (R² values) calculated from the resulting calibration curves were all ≥0.9994, indicating excellent linearity within the concentration range tested.

The LOD and LOQ, calculated in accordance with ACS guidelines [21], were 0.002 mg/L and 0.005 mg/L, respectively, for all the analytes investigated. The established quantification limits obtained in this study allow for precise measurements of these compounds, even at low concentrations that might be present in food samples. These limits are significantly below the total specific migration limit (SML(T)) set by Regulation (EC) No 10/2011 [11], which is 1.2 mg/kg and is expressed as the total sum of substances of N,N-bis(2-hydroxyethyl)alkyl (C8-C18) amines, where the difference between consecutive molecules in this group is the addition of a CH_2_ unit in their alkyl chains. The LODs obtained in this study are similar to those reported by Otoukesh et al. [5].

The precision and recovery of the extraction method were evaluated through spiking experiments conducted on food samples. The food samples were spiked with a mixed standard solution at three different concentrations over three consecutive days, resulting in a total of six measurements. Each analysis was performed in duplicate. To determine the exhaustiveness of the extraction process, food samples were subjected to several extraction steps. It was observed that performing a second extraction step resulted in recoveries that were less than 10% of those obtained in the first extraction. Therefore, it was concluded that the extraction of the food samples could be considered exhaustive after the initial extraction step. The results of the recovery (%) and intermediate precision expressed as relative standard deviation (RSD%) are shown in Table 3. The results reported provide evidence that the optimized method achieved acceptable recoveries that were in the range from 87% at the level of 0.02 µg/g for banana samples (S10) to 121% at the level of 0.1 µg/g for the sweet biscuit sample (S7). Regarding the intermediate precision values obtained, all of them were lower than 15%. These values obtained meet the method performance acceptability criteria [22].

### 3.4. Analysis of Food Packaging Materials using LC-MS/MS

It is crucial to note that this study encompasses the analysis of both sides of the material, including both the internal and external surfaces. First, the extracts of the packaging materials were analyzed using the SRM mode since this presents more sensitivity. This method allowed the positive confirmation of the presence of the four targeted chemical compounds with available analytical standards. Table 4 shows the concentration of the chemical compounds of interest in the extracts of the analyzed packaging materials. As can be seen, of the twelve samples analyzed, only four of them (S1, S2, S3 and S9) did not present quantifiable levels of any of the analytes of interest. The highest level of the tertiary amines was found in the extract corresponding to sample S12, with a concentration of 0.37 µg/dm^2^ for N,N-bis(2-hydroxyethyl)dodecylamine and 35.51 µg/dm^2^ for stearyldiethanolamine, while the highest concentrations of the secondary and primary amines were quantified in the extract of sample S10 (29.47 and 0.49 µg/dm^2^, respectively). It is interesting to remark that 2-(octadecylamino)ethanol and octadecylamine are not listed in Regulation 10/2011 for plastic materials for food contact [11]. Figure 2 shows LC-MS/MS chromatograms for the ethanol extract of S10 packaging. As can be seen, the four target compounds in the corresponding monitored MS/MS transitions were detected (see Appendix A).

These extracts, as well as the PEG-2 hydrogenated tallow amine, were also injected in full scan mode to try to identify other additives (tertiary amines) and possible impurities. Table 5 shows a summary of the compounds tentatively identified in the extracts of the packaging materials with their retention time, *m*/*z*, and their toxicity. All of them correspond to the ion [MH^+^] and the *m*/*z* ratios coincide with those reported in the literature [14,15], increasing the confidence in the identification process. Samples S1–S3 were not included in this table, since they did not present any of the chemical compounds.

The Cramer rules were utilized to evaluate the toxicological risk associated with these chemical compounds (Toxtree v3.1.0 software). This software employs a decision tree approach to predict the toxicological hazard based on the molecular structure of the compound. The classification is as follows: compounds considered to have low toxicity (class I), compounds classified as intermediate toxicity (class II), and compounds assigned high toxicity (class III) [23].

As can be seen in the table, the tertiary amine most frequently detected in the analyzed samples was N,N-bis(2-hydroxyethyl)octadecylamine, followed by N,N-bis(2-hydroxyethyl)hexadecylamine, while primary amines were only detected in PEG-2 hydrogenated tallow amine. All identified compounds had low toxicity (class I) except the secondary amines, 2-(octadecylamino)ethanol and 2-(hexadecylamino)ethanol which had high toxicity (class III). 2-(octadecylamino)ethanol has been identified in seven of the packaging materials analyzed and in the PEG-2 hydrogenated tallow amine, and as it has been previously commented, it is not included in the EU positive list for plastic food contact materials [11]. Moreover, they are predicted as likely to meet the criteria for category 1A or 1B carcinogenicity, mutagenicity, or reproductive toxicity by the European Chemicals Agency (ECHA) [24]. The primary and secondary amines found in the PEG-2 hydrogenated tallow amine may be present as residues or possible impurities resulting from the production process [9].

### 3.5. Analysis of Food Samples’ Migration Tests through LC-MS/MS and Dietary Exposure Assessment

The results of the analysis carried out on the food samples through LC-MS/MS in SRM mode were negative (below the detection limit) in all the samples and for the four analytes for which the standard is available, except in samples S8, S11 and S12 where stearyldiethanolamine was identified below the quantification limit. These extracts were also injected in full scan mode and some of the additives were identified. Thus, the dihydroxy alkylamines with C18, C17, C16 and C15 were detected in sample S8, while the dihydroxy alkylamines with C17, C16 and C15 were detected in sample S7. In contrast, only two dihydroxy alkylamines with C17 and C15 were detected in sample S10 and solely the dihydroxy alkylamine with C17 was identified in sample S11. Three dihydroxy alkylamines specifically C18, C17 and C16 were found in sample S12.

Regarding the migration tests, the S4 sample (packaging for surimi) was negative for the four analytes, while in the S5 sample (coffee capsules), the chemical compounds N,N-bis(2-hydroxyethyl) dodecylamine and stearyldiethanolamine were below the quantification limit. Our results were lower than those reported by Otoukesh et al. [5]. In the coffee capsules analyzed in that study, the short-chain dihydroxy alkylamines were detected, both in the migration of coffee and Tenax^®^, reaching concentration levels of 1310 µg/kg for Tenax^®^ and 1050 µg/kg for coffee, and exceeding the SML(T) recommended by the European Commission [11]. In a recent study, He et al. [25] assessed the migration of N,N-bis(2-hydroxyethyl) dodecylamine from polypropylene microwavable food containers in 10% ethanol (*v*/*v*), with the values reported ranging from not detected to 0.08 mg/kg.

Besides the analysis in SRM, the migration extracts were also injected in full scan mode, but no additives or possible impurities were detected.

As an integral part of the risk assessment process, the exposure to alkylamines found in the food samples was estimated by combining migration data and consumption data obtained from the Spanish national consumption survey ENALIA-2, specifically for the adult population. To calculate the dietary exposure, the GEMS/Food-EURO recommendations were followed. In cases where the values were below the limit of quantification (LOQ), the values were considered to be equal to LOQ/2 [26,27]. The mean exposure to stearyldiethanolamine ranged from 0.0005 to 0.0026 µg/kg bw/day (Table 6). Generally, the values estimated were low; nevertheless, it is noteworthy to observe that consumers are exposed to these chemicals through their diet from multiple origins (aggregate exposure) and, in addition, only some selected foods have been included in this study [28]. Very limited dietary exposure data of alkylamines migrated from food contact materials have been reported in the literature. In a recent study, He et al. [25] estimated the potential exposure to IAS and NIAS, including N,N-bis(2-hydroxyethyl) alkyl (C8-C18) amines transferred from microwavable plastic food containers. The average annual intake estimated was 55.15 mg, considering that take-out food was consumed once a day. The study was conducted in the 15 main cities in China.

## 4. Conclusions

The migration of N,N-bis(2-hydroxyethyl) alkyl (C8-C18) amines, as well as possible impurities from different packaging materials and coffee capsules, was investigated. Moreover, an industrial product specifically, PEG-2 hydrogenated tallow amine, was also analyzed. For that purpose, target and non-target approaches were carried out. N,N-bis(2-hydroxyethyl) alkyl amines, particularly C12, C13, C14, C15, C16, C17 and C18, were identified in almost all of the samples analyzed. Additionally, 2-(octadecylamino)ethanol was found in several samples; this substance is not listed in Regulation 10/2011, has been classified as high toxicity (class III) according to Cramer rules, and is predicted as likely to meet the criteria for category 1A or 1B carcinogenicity, mutagenicity, or reproductive toxicity (ECHA).

Regarding the migration tests, which were performed in foods and the food simulants Tenax and 20% ethanol (*v*/*v*), stearyldiethanolamine was determined at a concentration below the LOQ in tomato (S8), salty biscuits (S11), salad (S12) and in the food simulant Tenax (S5).

In an attempt to go one step further in the risk assessment, the exposure was estimated using the migration data and consumption data for the adult population obtained from ENALIA-2. The estimated values ranged from 0.0005 to 0.0026 µg/kg bw/day for the stearyldiethanolamine. Although, in general, they are low, it is important to point out that only some food products have been selected, and the combined exposure to multiple chemicals and from different sources have not been considered in this study. These aspects should be considered in future research for a complete risk assessment.

## Figures and Tables

**Figure 1 polymers-15-02656-f001:**
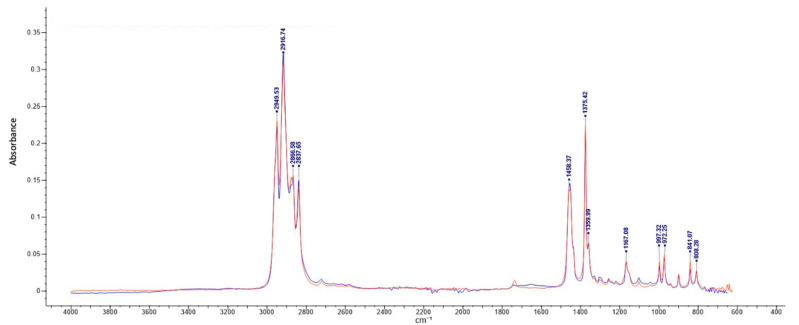
Infrared spectrum of the inner side of sample S4 (represented by the black line) compared to the first entry of the library (represented by the red line). Absorbance versus wavenumber (cm^−1^).

**Figure 2 polymers-15-02656-f002:**
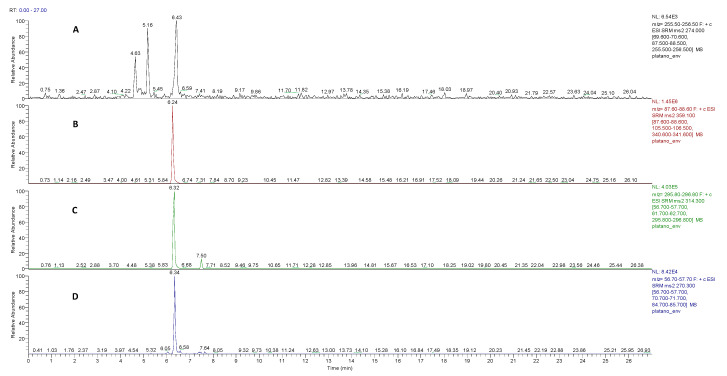
LC-MS/MS chromatograms for the ethanol extract of S10 packaging: (**A**) the transition for N,N-Bis(2-hydroxyethyl)dodecylamine (retention time: 5.16 min), (**B**) stearyldiethanolamine, (**C**) 2-(octadecylamino)ethanol and (**D**) octadecylamine.

**Table 1 polymers-15-02656-t001:** Information about the samples of this study.

Coding	Type of Food Sample	Type of Packaging Material		Thickness (µm)
		Internal Side	External Side	
S1	Gouda cheese	PE	-	52
S2	Alcoholic beverage	-	VP	47
S3	Orange juice	Polystyrene	PET	41
S4	Surimi	PP	PS	38
S5	Coffee	PP	PP	474
S6	Sweet biscuits	-	PP	25
S7	Sweet biscuits	PP	PP	23
S8	Tomato	PP	PP	18
S9	Fresh pasta	PE	Nylon (PA)	58
S10	Banana	PP	PP	20
S11	Salty biscuits	PP	PP	23
S12	Salad	PP	PP	25

PA: polyamide; PP: polypropylene; PE: polyethylene; PET: polyethylene terephthalate; PS: polyester; VP: copolymer vinyl chloride/vinyl acetate; -: not identified.

**Table 2 polymers-15-02656-t002:** Mass spectrometry conditions for the selected analytes.

Compound	Chemical Structure	CAS Number	Parent Ion	Product Ion *	Collision Gas Energy (V)
N,N-Bis(2-hydroxyethyl) Dodecylamine	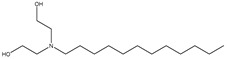	1541-67-9	274	88	22
256	22
Stearyldiethanolamine	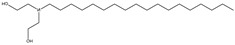	10213-78-2	359.1	88.1	28
341.1	22
2-(octadecylamino)ethanol	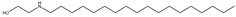	31314-15-5	314.3	296.3	20
57.2	26
Octadecylamine	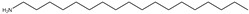	124-30-1	270.3	57.2	22
71.2	16

* The first product ion was used for quantification and the second for identification purposes.

**Table 3 polymers-15-02656-t003:** Results of method validation parameters.

Compound	Retention Time (min)	R^2^	Equation	Sweet Biscuits (S7)						Banana (S10)					
				Recovery (%)			Intermediate Precision (RSD%)			Recovery (%)			Intermediate Precision (RSD%)		
				0.02 µg/g	0.1 µg/g	0.2 µg/g	0.02 µg/g	0.1 µg/g	0.2 µg/g	0.02 µg/g	0.1 µg/g	0.2 µg/g	0.02 µg/g	0.1 µg/g	0.2 µg/g
N,N-Bis(2-hydroxyethyl) dodecylamine	5.16	0.9996	Y = 9204.9x − 59080	113	121	117	3	12	4	104	119	116	15	6	5
Stearyldiethanolamine	6.24	0.9994	Y = 4117.7x − 8353.7	106	108	114	9	9	6	87	105	99	14	8	11
2-(octadecylamino)ethanol	6.32	0.9994	Y = 1469.8x + 12173	108	103	107	8	9	8	105	104	117	10	7	5
Octadecylamine	6.34	0.9994	Y = 11348x − 84654	101	88	97	12	15	12	98	109	90	10	14	8

**Table 4 polymers-15-02656-t004:** Concentration of the chemical compounds of interest in the extracts of the analyzed packaging materials using LC-MS/MS (µg/dm^2^).

	N,N-Bis(2-hydroxyethyl) Dodecylamine	N,N-Bis(2-hydroxyethyl) Octadecylamine	2-(Octadecylamino)ethanol	Octadecylamine
S1	nd	nd	nd	nd
S2	nd	nd	nd	nd
S3	nd	nd	nd	nd
S4	nd	0.99	12.45	<LOQ
S5	nd	6.17	0.56	nd
S6	<LOQ	14.71	21.60	<LOQ
S7	0.30	31.95	3.24	<LOQ
S8	0.37	8.14	1.34	nd
S9	nd	<LOQ	nd	nd
S10	0.24	29.70	29.47	0.49
S11	nd	14.78	2.26	<LOQ
S12	0.37	35.51	18.03	<LOQ

nd: not detected.

**Table 5 polymers-15-02656-t005:** Compounds tentatively identified in the extracts of the packaging materials and their toxicity according to the Cramer rules.

Proposed Compound	*m*/*z*	CT	Retention Time (min)	PEG-2 Hydrogenated Tallow Amine	S4	S5	S6	S7	S8	S9	S10	S11	S12
N,N-bis(2-hydroxyethyl)dodecylamine *	274	I	5.16	X				X	X				X
N,N-bis(2-hydroxyethyl)tridecylamine	288	I	5.33	X				X	X			X	
N,N-bis(2-hydroxyethyl)tetradecylamine	302	I	5.53	X			X	X	X		X	X	X
N,N-bis(2-hydroxyethyl)pentadecylamine	316	I	5.70	X	X			X	X		X	X	
N,N-bis(2-hydroxyethyl)hexadecylamine	330	I	5.87	X	X	X	X	X	X		X	X	X
N,N-bis(2-hydroxyethyl)heptadecylamine	344	I	6.02	X			X	X	X		X	X	X
N,N-bis(2-hydroxyethyl)octadecylamine *	358	I	6.24	X	X	X	X	X	X	X	X	X	X
2-(octadecylamino)ethanol *	314	III	6.32	X	X		X	X	X		X	X	X
2-(hexadecylamino)ethanol	286	III	5.96	X									
Octadecylamine *	270	I	6.34	X									
Hexadecylamine	242	I	5.99	X									

* confirmed with standard; CT: Cramer toxicity.

**Table 6 polymers-15-02656-t006:** Estimated dietary exposure in Spanish adult population to stearyldiethanolamine.

Sample	Consumption	Dietary Exposure (µg/kg bw/day)		
		Mean	P50	P95
S8	1.04	0.0026	0.0019	0.0079
S11	0.19	0.0005	0.0004	-
S12	1.05	0.0026	0.0021	0.0060
S8	1.04	0.0026	0.0019	0.0079

## Data Availability

The data presented in this study are available in [Migration of dihydroxy alkylamines and their possible impurities from packaging into foods and food simulants: Analysis and safety evaluation].

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
