# Peer review of "Migration of Dihydroxy Alkylamines and Their Possible Impurities from Packaging into Foods and Food Simulants: Analysis and Safety Evaluation"

_polymers, 2023, doi:10.3390/polym15122656_

Round 1
Reviewer 1 Report
Such research is meaningful, but in absence of novelty, so the accuracy of the data is very important. The paper is written well, but I still have some questions and comments. If they are answered reasonably, the paper can be published in this journal.
(1) Why did IR characterization? What’s the difference between the two spectra in Figure 1? If the authors want to identify different packaging materials, more IR spectra of various food packaging materials should be tested.
(2) From Figure 2A, why there are three peaks? Which one is N,N-Bis(2-hydroxyethyl)dodecylamine ? Please label them clearly or claim in the figure caption.
(3) Figure 2B-2D, we can see the retention times are very close, if the sample contained the three targets together, how to perform the qualification and quantification ? please provide the LC-MS/MS chromatograms of the 12 samples, particularly S4-S8, S10-S12, and explain how to get the quantification results in Table 4.
(4) References’ format should be revised.
(5) Some grammar errors should be checked and corrected, e.g. Line 148, “To carried out recovery tests,”
Author Response
Ms. Christine Liu Assistant Editor Polymers
Dear Editor,
We appreciate the contribution that your comments and suggestions have on the improvement of the manuscript.
In order to revise the manuscript polymers-2451489 entitled: “Migration of dihydroxy alkylamines and its possible impurities from packaging into foods and food simulants: Analysis and safety evaluation” the following comments and changes were made in accordance with the reviewer’s and editor guidelines.
Reviewer 1:
Such research is meaningful, but in absence of novelty, so the accuracy of the data is very important. The paper is written well, but I still have some questions and comments. If they are answered reasonably, the paper can be published in this journal.
(1) Why did IR characterization? What’s the difference between the two spectra in Figure 1? If the authors want to identify different packaging materials, more IR spectra of various food packaging materials should be tested.
IR characterization of the available food packaging materials was carried out to try to identify the type of polymer present and try to establish a relationship between the type of polymer and the presence/absence of dihydroxy alkylamines (as can be observed with the PP polymer and coincides with what was reported in references 5 and 14).
In Figure 1, the red spectrum is the first entry proposed by the commercial library (in this case it is the spectrum corresponding to the PP polymer) and the black spectrum is the spectrum obtained for the S4 samples on the internal side.
The objective of this work was to develop an analytical method to determine dihydroxy alkylamines and, at the same time, test it by doing a small screening between samples provided by the industry and samples purchased in supermarkets.
(2) From Figure 2A, why there are three peaks? Which one is N,N-Bis(2-hydroxyethyl)dodecylamine ? Please label them clearly or claim in the figure caption.
In Figure 2A more peaks can be observed (which were not identified) because it is a real sample where there are surely more compounds.
The peak that corresponds to N,N-Bis(2-hydroxyethyl)dodecylamine is the one that comes out at the retention time reported in Table 3, specifically at 5.16 min. So that there is no confusion, it was indicated in the figure caption.
(3) Figure 2B-2D, we can see the retention times are very close, if the sample contained the three targets together, how to perform the qualification and quantification ? please provide the LC-MS/MS chromatograms of the 12 samples, particularly S4-S8, S10-S12, and explain how to get the quantification results in Table 4.
Figure 2 is an example of the case that the reviewer proposes, where we can see that all the target compounds are present in the sample S10. Indeed, with liquid chromatography, it is not enough to separate the 3 compounds. It is for this reason that detection by tandem mass spectrometry (MS/MS) was selected, which allows separation based on their mass/charge ratio (m/z). In addition, a precursor ion is fragmented into product ions and the MS/MS instrument identifies the compound of interest by its unique constituent parts.
To carry out the identification and quantification of the target compounds, two transitions for each compound were selected and they are presented in Table 2.
The chromatograms of the positive samples (S4-S8, S10-S12) are provided in supplementary material (see Figure S1-S7).
Quantification of these compounds was conducted using an external calibration curve. The chromatographic peaks' areas in the selected transitions for quantification (as listed in Table 2 and as shown in the example of Figure 2 for sample S10) were plotted against the corresponding concentrations of the standard solutions to generate the calibration graphs.
(4) References’ format should be revised.
Following the recommendation of the reviewer, the references were revised according to the Polymers template.
(5) Some grammar errors should be checked and corrected, e.g. Line 148, “To carried out recovery
tests,”
According to this suggestion of the reviewer, the text was checked and corrected.
We hope all explanations and changes will meet your approval. Hope to hearing from you as soon as possible.
Yours faithfully,
Ana Rodríguez Bernaldo de Quirós

Reviewer 2 Report
L 36: PVC is hardly used anymore.
L 38-39: Too general. The plastic industry rapidly develops so it is not clear if these substances are still used so much in food packaging.
L 51-51: According to IUPAC PE-LD, PE-HD etc.
L 32-91: Can the authors provide a proof that ‘dihydroxy alkylamines’ are really used in food packaging? Sometimes such things are not so clear, than it should be written a ‘obvious, without proof’.
Table 1: Is it possible to specify the polymers better, e.g., is the PE a PE-LD. Different grades of the same polymer behave differently.
Table 4: Is known why the substance is sometimes used and why sometimes not at specific examples?
Author Response
Ms. Christine Liu Assistant Editor Polymers
Dear Editor,
We appreciate the contribution that your comments and suggestions have on the improvement of the manuscript.
In order to revise the manuscript polymers-2451489 entitled: “Migration of dihydroxy alkylamines and its possible impurities from packaging into foods and food simulants: Analysis and safety evaluation” the following comments and changes were made in accordance with the reviewer’s and editor guidelines.
Reviewer 2:
L 36: PVC is hardly used anymore.
According to this suggestion of the reviewer, PVC was removed from the text.
L 38-39: Too general. The plastic industry rapidly develops so it is not clear if these substances are still used so much in food packaging.
The authors of the present work analyzed plastic food packaging in other previous works where they found these families of additives in real samples (antioxidants, slip agents, plasticizers, lubricants, light stabilizers, among others). The reviewer can check it in the following references:
Lestido-Cardama, A., Rodríguez Bernaldo de Quirós, A., Bustos, J., Lomo, M. L., Paseiro Losada, P., & Sendón, R. (2020). Estimation of dietary exposure to contaminants transferred from the packaging in fatty dry foods based on cereals. Foods, 9(8), 1038.
Lestido-Cardama, A., Sendón, R., Bustos, J., Lomo, M. L., Paseiro Losada, P., & Rodríguez Bernaldo de Quirós, A., (2020). Dietary exposure estimation to chemicals transferred from milk and dairy products packaging materials in Spanish child and adolescent population. Foods, 9(11), 1554.
Regarding these substances, namely dihydroxy alkylamines, they are still being used, in fact EFSA recently launched a call requesting migration data for these substances.
L 51-51: According to IUPAC PE-LD, PE-HD etc.
Following the recommendation of the reviewer, the names were changed.
L 32-91: Can the authors provide a proof that ‘dihydroxy alkylamines’ are really used in food packaging? Sometimes such things are not so clear, than it should be written a ‘obvious, without proof’.
The present work, as well as the cited references (see references 5, 10, 14, 15), are proof that the ‘dihydroxy alkylamines’ are used in food packaging. Moreover, they are permitted for use in plastic food materials according to Regulation 10/2011.
Table 1: Is it possible to specify the polymers better, e.g., is the PE a PE-LD. Different grades of the same polymer behave differently.
The authors agree with the reasoning of the reviewer that different grades of the same polymer could behave differently. However, unfortunately, the available IR spectral library did not allow us to specify the best type of polymers.
Table 4: Is known why the substance is sometimes used and why sometimes not at specific examples?
The IR characterization of food packaging materials allow us to establish a relationship between the type of polymer and the presence/absence of dihydroxy alkylamines. As can be observed, when the packaging materials were identified as PP polymer (samples: S4-S8, S10-S12), some dihydroxy alkylamines were detected, which coincides with what was reported in references 5 and 14.
With our results, it can also be deduced that tertiary amine N,N-Bis(2-hydroxyethyl)octadecylamine is more used in food packaging than N,N-Bis(2-hydroxyethyl)dodecylamine, and that the secondary amine 2-(octadecylamino)ethanol was only detected in those samples in which the tertiary amine N,N-Bis(2- hydroxyethyl)octadecylamine was present.
We hope all explanations and changes will meet your approval. Hope to hearing from you as soon as possible.
Yours faithfully,
Ana Rodríguez Bernaldo de Quirós

Reviewer 3 Report
This work summarizes the Migration of dihydroxy alkylamines and its possible impurities from packaging into foods and food simulants: Analysis and safety evaluation. The manuscript structure is well organized, and the recent work is interesting for packaging applications. I am recommending this manuscript for publication. However, some minor comments should be supplied before publication. For example, the author should add the abbreviation name of the OECD. In Figure 1, the author should add the wavenumber corresponding to the peak, and the caption of the x-axis and y-axis should be added. The author should add more discussion details about Figure 2.
Author Response
Ms. Christine Liu
Assistant Editor
Polymers
Dear Editor,
We appreciate the contribution that your comments and suggestions have on the improvement of the manuscript.
In order to revise the manuscript polymers-2451489 entitled: “Migration of dihydroxy alkylamines and its possible impurities from packaging into foods and food simulants: Analysis and safety evaluation” the following comments and changes were made in accordance with the reviewer’s and editor guidelines.
Reviewer 3:
This work summarizes the Migration of dihydroxy alkylamines and its possible impurities from packaging into foods and food simulants: Analysis and safety evaluation. The manuscript structure is well organized, and the recent work is interesting for packaging applications. I am recommending this manuscript for publication. However, some minor comments should be supplied before publication. For example, the author should add the abbreviation name of the OECD. In Figure 1, the author should add the wavenumber corresponding to the peak, and the caption of the x-axis and y-axis should be added. The author should add more discussion details about Figure 2.
Following the recommendation of the reviewer, the abbreviation has been included. Moreover the Figure 1 has been modified according the reviewer’s suggestion and the Figure 2 commented on.
We hope all explanations and changes will meet your approval.
Hope to hearing from you as soon as possible.
Yours faithfully,
Ana Rodríguez Bernaldo de Quirós

Round 2
Reviewer 1 Report
Every questions are answered well.